# Estimating Connectivity of Hard Clam (*Mercenaria mercenaria*) and Eastern Oyster (*Crassostrea virginica*) Larvae in Barnegat Bay

Jacob D. Goodwin [1,2,*], Daphne M. Munroe [1], Zafer Defne [3] 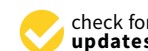, Neil K. Ganju [3] and James Vasslides [4]

[1]   Haskin Shellfish Research Laboratory, Rutgers University, Port Norris, NJ 08349, USA; dmunroe@hsrl.rutgers.edu
[2]   Department of Biology, Woods Hole Oceanographic Institution, Woods Hole, MA 02543, USA
[3]   U.S. Geological Survey, Woods Hole, MA 02543, USA; zdefne@usgs.gov (Z.D.); nganju@usgs.gov (N.K.G.)
[4]   Barnegat Bay Partnership, College Drive, Toms River, NJ 08753, USA; jvasslides@ocean.edu
[*]   Correspondence: oysterjake@aol.com or jgoodwin@whoi.edu; Tel.: +1-410-330-1502

**Abstract:** Many marine organisms have a well-known adult sessile stage. Unfortunately, our lack of knowledge regarding their larval transient stage hinders our understanding of their basic ecology and connectivity. Larvae can have swimming behavior that influences their transport within the marine environment. Understanding the larval stage provides insight into population connectivity that can help strategically identify areas for restoration. Current techniques for understanding the larval stage include modeling that combines particle attributes (e.g., larval behavior) with physical processes of water movement to contribute to our understanding of connectivity trends. This study builds on those methods by using a previously developed retention clock matrix (RCM) to illustrate time dependent connectivity of two species of shellfish between areas and over a range of larval durations. The RCM was previously used on physical parameters but we expand the concept by applying it to biology. A new metric, difference RCM (DRCM), is introduced to quantify changes in connectivity under different scenarios. Broad spatial trends were similar for all behavior types with a general south to north progression of particles. The DRCMs illustrate differences between neutral particles and those with behavior in northern regions where stratification was higher, indicating that larval behavior influenced transport. Based on these findings, particle behavior led to small differences (north to south movement) in transport patterns in areas with higher salinity gradients (the northern part of the system) compared to neutral particles. Overall, the dominant direction for particle movement was from south to north, which at times was enhanced by winds from the south. Clam and oyster restoration in the southern portion of Barnegat Bay could serve as a larval supply for populations in the north. These model results show that coupled hydrodynamic and particle tracking models have implications for fisheries management and restoration activities.

**Keywords:** bivalve connectivity; larval transport; modeling; retention clock; RCM; ROMS; LTRANS; Barnegat Bay; hard clam; eastern oyster

## 1. Introduction

Shellfish provide economic, cultural, and ecological value by supporting local fisheries [1], creating habitat [2], and providing ecosystem services (e.g., water filtration [3]). Increasingly, shellfish restoration has been proposed as a mechanism to improve water quality in eutrophic estuaries [3,4], and to help mitigate shoreline loss through stabilization [5]. As much as we are commonly interested

in the adult shellfish stages, it is the dispersive planktonic larval stage that influences population distributions because the juvenile and adult stages are sessile [2,6]. It is important to understand this stage to effectively manage shellfish. Some research has been conducted on the crucial planktonic stage that governs the recruitment patterns in a shellfish population [7,8]. Understanding the abundance and distribution patterns of shellfish larvae requires a large number of samples over time and space [9–11], but advances in larval transport modeling can provide insight into the transient life stage of these ecologically and economically important shellfish [8,12–14].

Commercially important shellfish have declined sharply in U.S. estuaries and bays from Maine to North Carolina and some of these declines have been linked to shifts in the North Atlantic Oscillation [15]. The lack of continuous survey records of abundance and distribution of shellfish in Barnegat Bay Little Egg Harbor (BBLEH) makes inference about possible population trends over time difficult [16]. Hard clam surveys performed by the New Jersey Department of Environmental Protection (NJDEP) Bureau of Shellfisheries have shown that wild adult hard clam populations have been in decline [17]. Although data is limited the decline between the early 1980s and 2000s have been reported as 36% decline for oysters and 83% for clams in Barnegat Bay [15]. Recently, community groups interested in conservation and enhancement of the estuary (e.g., the Barnegat Bay Partnership) have identified shellfish stock enhancement as a key strategic priority because shellfish are key ecosystem components.

Spawning adult clams and oysters produce planktonic larvae that can remain in the water column for several weeks and be dispersed within the system [2,18]. Understanding the patterns of larval dispersal, transport, and connectivity within a local system is an important component in strategic planning of shellfish restoration and enhancement projects as well as prioritizing protections of existing shellfish beds. Larval dispersal refers to the spread of larvae from a spawning source to a settlement site [19]. Larval transport is defined as the horizontal translocation of a larva between two points, and connectivity is defined as larval transport from one population to another [19].

Coupled bio-physical modeling is often used to understand population connectivity of marine organisms with planktonic stages [12–14]. The Regional Ocean Modeling Systems (ROMS) [20], coupled with particle tracking models (e.g., Larval TRANSport Lagrangian model; LTRANS) have been used to infer dispersal and connectivity of planktonic particles in coastal systems [12–14,21]. Models similar to these have been used to calculate probability-based connectivity for different species in a variety of systems [8,22,23]. The models used have uncertainties that include the variability in seasonal circulation, interaction of particles with boundaries and larval swimming behavior but they can still provide valuable insight into overall transport patterns.

Some models only display output describing connectivity among regions for a specific time or condition, but these results often do not reflect variability in connectivity over larval development time. Defne et al. [21] introduced a retention clock matrix (RCM) that allows simultaneous spatio-temporal analysis of coupled biological and physical model output. The RCM concept, displayed for abiotic passive particles previously, can also display changes in connectivity across a range of pelagic larval durations (PLD, the length of time a larva spends in the water column after hatching or spawning, before settlement). Ranges in the PLD of marine organisms can vary by weeks to months within species [2,22] and it may be useful for managers to display variable connectivity patterns and PLDs to better understand connectivity of a particular species in a system of interest. Traditional connectivity matrices offer one snapshot in time but the RCM allows multiple dispersal times to be viewed on the same diagram (Figure 1). The diagram consists of a pie chart for each source–destination pair and each slice represents the connectivity of the pair in time. A novel concept is also introduced in our methods to compare connectivity patterns illustrated using the difference between RCMs or DRCM (Difference Retention Clock Matrix).

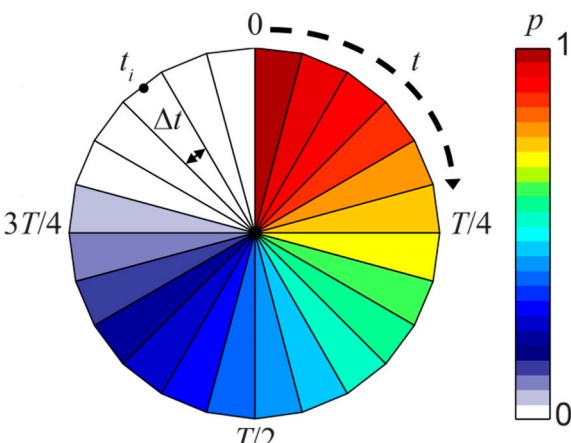

**Figure 1.** An example of a retention clock, adapted from 21. Time progresses clockwise from 0 to a preset maximum time *T* (in subsequent figures, *T* = 22 days) with a time step of *t* (*t* = 2 days in all subsequent figures). Color scales represent the ratio of particles (*P*) initiated in the source polygon which are observed in a given destination polygon.

Two economically and ecologically important species in the U.S. are hard clams (*Mercenaria mercenaria*) and eastern oysters (*Crassostrea virginica*). Clams are particularly important serving as "the social fabric" of the BBLEH region of New Jersey [24,25], and together with oysters, contribute directly and indirectly to the four billion dollar economic value of the system [26].

The main objective of this research was to model connectivity in BBLEH for clams and oysters. We demonstrate the utility of RCMs in our modeled system by using a physical hydrodynamic model (ROMS), coupled with a particle tracking behavior model (LTRANS) to predict patterns of connectivity for hard clams and eastern oysters in BBLEH estuary.

## 2. Methods

### 2.1. Barnegat Bay Little Egg Harbor (BBLEH)

The BBLEH estuary is a back-barrier estuary approximately 70 km long, on the coast of New Jersey, USA (Figure 2). The system is relatively shallow (1.5 m) with restricted exchange through the Manasquan Inlet and Point Pleasant canal (north), the Barnegat Inlet (central), and Little Egg Inlet (south). The northern part of the bay is tidally less energetic with more stratification than the southern half, and has less volume exchange with the open ocean based on a previous ROMS model [27].

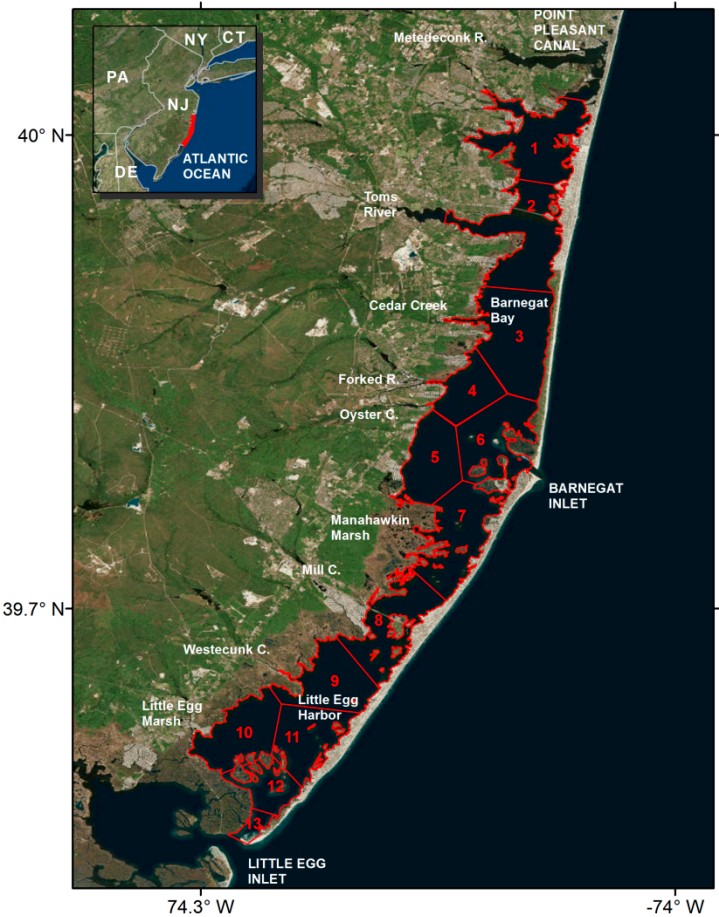

**Figure 2.** Barnegat Bay and Little Egg Harbor (BBLEH), New Jersey, USA with particle release points centered within each of the numbered 1–13 polygons (outlined in red). Manasquan Inlet (not shown) is to the north of Point Pleasant Canal.

*2.2. Modeling*

Particle dispersal was simulated for clam and oyster larvae in BBLEH estuary (Figure 2). A larval transport model incorporated the 3-D ROMS hydrodynamic model of the bay [27] with the LTRANS behavior and particle-tracking-model [14] to simulate movement of particles in the system. The hydrodynamic model had a skill score that ranged from very good to excellent in predicting water levels and tidal discharges according to the Brier Skill Score [27]. Physical model output was only available for the spring, prior to the period during which shellfish spawning is typical. From this model output, we selected a window of time that had southerly wind conditions similar to those commonly seen during a summer spawning period to use as input for the larval transport model (Figure S1). A ROMs simulation period of two months showed that the general circulation patterns of neutral particles hold over a wide variety of conditions in this estuary [27].

The model used in this study simulated the physical parameters of the bay using the ROMS module of the Coupled Ocean Atmospheric Wave Sediment Transport System (COAWST) [21,28]. The hydrodynamic model was forced by winds, tides, and freshwater input for a three month period in BBLEH (Figure S2). At the western landward boundary point, sources of freshwater were identified in accordance with U.S. Geological Survey streamflow measurements [29] and a radiation boundary condition that allowed tidal energy to propagate landward. On the eastern seaward side, tidal water level and velocity amplitudes were applied (ADCIRC database for the North Atlantic [30]) and supplemented by subtidal water level and subtidal barotropic velocities (ESPreSSO model for the Mid-Atlantic Bight [31]. Meteorological forcing from North American Mesoscale Model [32] was

applied at the ocean–atmospheric interface. The bulk flux parameterization routine was used with 3-h wind velocity, air pressure, long—and shortwave radiation, relative humidity, and rain inputs. More detail on model forcing at lateral boundaries and atmospheric forcing are described in [21,27]. The model had seven evenly distributed vertical layers, a horizontal resolution between 40–200 m, and a time step of five seconds. Larval simulations focused on a time period of 22 days using model output from ROMS starting 15 March 2012. Model output was used to calculate the maximum vertical salinity gradient (halocline) over the 22 day time period because salinity gradients have a large effect on larval swimming behavior [2,14,18], and are used in the larval transport model as a behavior cue [33].

The LTRANS model used the velocity, density, salinity, and vertical diffusivity output from the ROMS model for particle tracking simulations [33]. The model was designed to predict movement of particles based on advection, turbulence (both horizontal and vertical), and larval swimming behavior. The 3-D water properties of the model [34], and the larval swimming behavior is defined by the behavior sub-model [33]. A random displacement model [35,36] was used to simulate sub-grid scale turbulent particle motion. Further explanation of the LTRANS model and the code can be found in [14].

The LTRANS behavior sub-model (described in detail in [33]) assigns each particle a vertical sinking, floating, or swimming velocity that responds to the fluid environment at each time step, thereby regulating the vertical motion. Three categories of simulations were performed using neutral, clam, and oyster particle behaviors (Table 1). Neutral particles were programmed with no behavior, thus the particle path was a result of advection and turbulence components of the hydrodynamic model.

A simulation was run for clam particles with a pelagic larval duration (PLD) of 22 days. Two aspects of behavior were controlled: swimming speed (linear) and swimming direction (varies with age and vertical salinity gradient) (Table 1). Clam behavior was implemented using the same parameters as for oysters in [14], but with different swimming speeds, larval duration, and threshold salinity gradient that cued swimming behavior (Table 1). Clam swimming behavior was based on a review of previous studies on both hard clams *Mercenaria mercenaria* [18] and surf clams *Spisula solidissima* [37]. Based on these reviews, clam particles were programmed to slow down from veliger to pediveliger stage linearly over time. Particles 0–1 days old were assumed fertilized gametes and assigned no swimming behavior based on [18]. Simulated clam particles began swimming at day 1 after release, and swimming speeds decreased linearly by age ($y = -6.15x + 0$, range 0–1.3 mm s$^{-1}$) between veliger and pediveliger stages [18]. The maximum swimming speed was multiplied by a number drawn from a uniform random distribution (0–1) to simulate random variations in the movements of individual clam larvae (after 14). At the pediveliger stage (day 8) swimming speeds were set at 0.49 mm s$^{-1}$ (from *Spisula solidissima* [37]) with no random component added. The vertical swimming direction of particles varied based on age and salinity gradient encountered. Particles between day 0.5 and day 1.5 were randomly assigned upward or downward swimming at each time step with a high probability (90%) of swimming upward. After day 1.5, if no halocline was present at the particle's location, the particle was assigned swimming probability that was a linear function of age resulting in a gradual shifting toward the bottom over time [33]. If a vertical salinity gradient equal to or in excess of the defined threshold for clams ($\geq$5.0 psu m$^{-1}$, [18] was present at that particle's location, the particle was assigned an 80% chance of swimming upward during the veliger 1–7 d stage, thereby retaining them above the salinity gradient prior to day 8.

**Table 1.** Parameterized behavior used in LTRANS to define both clam and oyster response to the environment in the model. Salinity gradient threshold (max vertical salinity gradient over 1 m) that cues larval behavior, age swimming begins, age of veliger stages, age of pediveliger stages, and swimming speeds at veliger and pediveliger stages for both clam and oyster sub-behaviors in the model.

|  | Clam | Oyster |
| --- | --- | --- |
| Salinity gradient threshold | 5 | 1.2 |
| Age swimming begins (days) | 1 | 1 |
| Age veliger (days) | 1–7 | 1–13 |
| Age pediveliger (days) | 8 | 14 |
| Swim speed veliger (mm s$^{-1}$) | 1.30–0.49 | 0.50–3.00 |
| Swim speed pediveliger (mm s$^{-1}$) | 0.49 | 3.00 |

A simulation was run for oyster larvae using a PLD of 22 days. The vertical swimming direction of particles varied based on age and salinity gradient encountered. Oyster particle behavior followed the parameterization for *C. virginica* in LTRANSv2 [33], with the exception of the response to the vertical salinity gradient. In the previous implementation, each particle responds to a vertical salinity gradient set at 1 psu m$^{-1}$ [14]. Our implementation used an updated response to a vertical salinity gradient set at 1.2 psu m$^{-1}$ [9]. Oyster particles from 0 to 0.5 days old were assumed fertilized gametes and did not swim. Particles at day 0.5 began to swim at 0.5 mm s$^{-1}$ increasing linearly to 3 mm s$^{-1}$ until they reached the pediveliger stage at day 14 (y = 3 x − 1.6). The maximum swimming speed was multiplied by a number drawn from a uniform random distribution (0–1) to simulate random variations in the movements of individual oyster larvae. At the pediveliger stage, the random component was dropped and larvae swimming speeds were 3 mm s$^{-1}$. The vertical swimming direction of oyster particles between day 0.5 to day 1.5 were randomly assigned upward or downward swimming at each time step with a high probability (90%) of swimming upward. With no encounter of a halocline, particles are assigned swimming probability that was a linear function of age resulting in a gradual shifting toward the bottom over time [33]. However, if the vertical salinity gradient at the particle's location equaled or exceeded the threshold (≥1.2 psu m$^{-1}$), the oyster particle was assigned an 80% chance of swimming upward during the veliger 1–13 d stage, thereby retaining them above the salinity gradient.

Before release, to represent spawning, all particles were positioned uniformly at the bottom layers and the center of each of the 13 polygons that were defined by using a Thiessen polygon approach [38] so that modeled results could encompass the entire estuary. This approach carved out various shaped polygons with different areas throughout the system. In addition to capturing broader trends, releasing particles from the center of each polygon allowed us to compare results with previous modeled efforts in this system [27]. In these simulations, particles were released on 15 March 2012 because this period had wind conditions that were similar to conditions typical of the spawning season in BBLEH. The temperature conditions may not have reflected spawning conditions; however, larval particle behavior in these simulations was not affected by temperature. Each simulated behavior (neutral, clam, oyster) was repeated by releasing 1000, 2500, and 5000 particles from the center of each of the 13 polygons (13,000; 32,500; and 65,000 total particles released during each simulation respectively). In total, nine particle tracking model runs were conducted from the same hydrodynamic model output.

*2.3. Retention Clock Matrix*

Retention clock matrices (RCMs) were used to illustrate connectivity output from the bio-physical coupled model simulations. Detailed description of the method and a pseudo code are provided in [21]. In summary, if a single release of particles is defined as

$$p = \frac{N(t_i)}{N_0} \tag{1}$$

where $p$ is the proportion of particles retained at a given time, $N(t_i)$ represents the number of particles in the domain (polygon in this study) at time $t_i$ and $N_0$ is the total number of particles initially released. A retention clock describes the release event (Equation (1)) as a circular clock that spans the entire time scale ($T$) of interest with the temporal resolution of $\Delta t$, and with $p$ ranging from 0 to 1 (Figure 1). This allows the concentration of particles $p$ at each point around the retention clock $t$ to be viewed together.

A maximum time scale of 22 days was used for all model scenarios. Each slice on all retention clocks represents 2 days. We assumed that connectivity was insignificant if it did not satisfy a minimum threshold of at least 1% of particles released from the source being dispersed to the destination polygon. Therefore, clock slices shown in color indicate connectivity between the point released and the destination polygon, whereas no clock is plotted for a particular source–destination combination if connectivity is less than 1%. This assumption enabled highlighting general trends rather than marginal connectivity. In all RCMs the $y$-axis represents the 13 points from which particles were released (source polygon), while the $x$-axis represents where the particles ended up (destination polygon), and the strength of connectivity between each possible combination of source-destination is shown with a color bar (Figure 1). Connectivity between a release point and destination polygon is denoted by the corresponding $y$- and $x$-axis label pairs (e.g., see later results for RCM analysis). For example, the transport of particles released from source 2 (on $y$-axis) that arrived at destination 3 (on $x$-axis) is indicated by (s2d3).

A Difference Retention Clock Matrix (DRCM) was used to illustrate the change in connectivity between different particle behavior simulations, and to assess output sensitivity to varying the numbers of particles released. The DRCM is defined as

$$DRCM_{2-1}(t_i, s, d) = RCM_2(t_i, s, d) - RCM_1(t_i, s, d) \tag{2}$$

where $RCM_1(t_i, s, d)$ is number of particles from source $s$ at destination $d$ at time $t_i$ for scenario one (i.e., the slice of the RCM that corresponds to $t_i$), and $RCM_2$ is likewise for the second scenario. Hence, DRCM shows the difference in connectivity between the two scenarios and has the same structure with the compared RCMs. For example, the DRCM between 5000 particles released from each site and one with 1000 is denoted by $DRCM_{5000-1000}$. Positive and negative values of DRCM indicate stronger and weaker connectivity in the second scenario, respectively relative to the first one. To illustrate the greatest difference between two scenarios during the simulation duration, a $DRCM_{max}$ was calculated as

$$DRCM_{max} = Max\left(\left|DRCM_{1-2}(t_i, s, d)\right|, \; for \; t_i = t_1 \; to \; T\right) \tag{3}$$

which is the maximum of the absolute value of the slices of the DRCM. The $DRCM_{max}$ can be used to quantify the maximum absolute difference between RCMs, whereas DRCM can be used to illustrate the difference at a given point in time and space.

Additionally, an estimate of the relative degree of connectivity of each polygon to the rest of the system was determined by summing the number of other polygons that received particles from a given source polygon. Discrete PLDs (rather than variable PLDs as depicted in RCMs) of 8 days for clams and 14 days for oysters were used for these relative connectivity summations. These discrete PLDs were chosen based on mean literature values of PLDs for these particular species in this system based on their observed transitions from veliger to pediveliger and competence to settle [2,18].

## 3. Results

Vertical salinity gradients during the simulated period were stronger in the northern stations and reached the clam model behavior threshold of 5 psu m$^{-1}$, thereby initiating clam behavior in some polygons (Figure 3). The oyster larval behavior threshold of 1.2 psu m$^{-1}$ was more commonly reached baywide (Figure 3). Polygons in the southern portion of the estuary tended to have a negligible salinity gradient compared to the polygons in the north (Figure 3). Within the first 4 days of simulation, winds

were blowing from the northeast with speeds up to 5 m per second followed predominately by winds from the south with the same speeds (Figure S1).

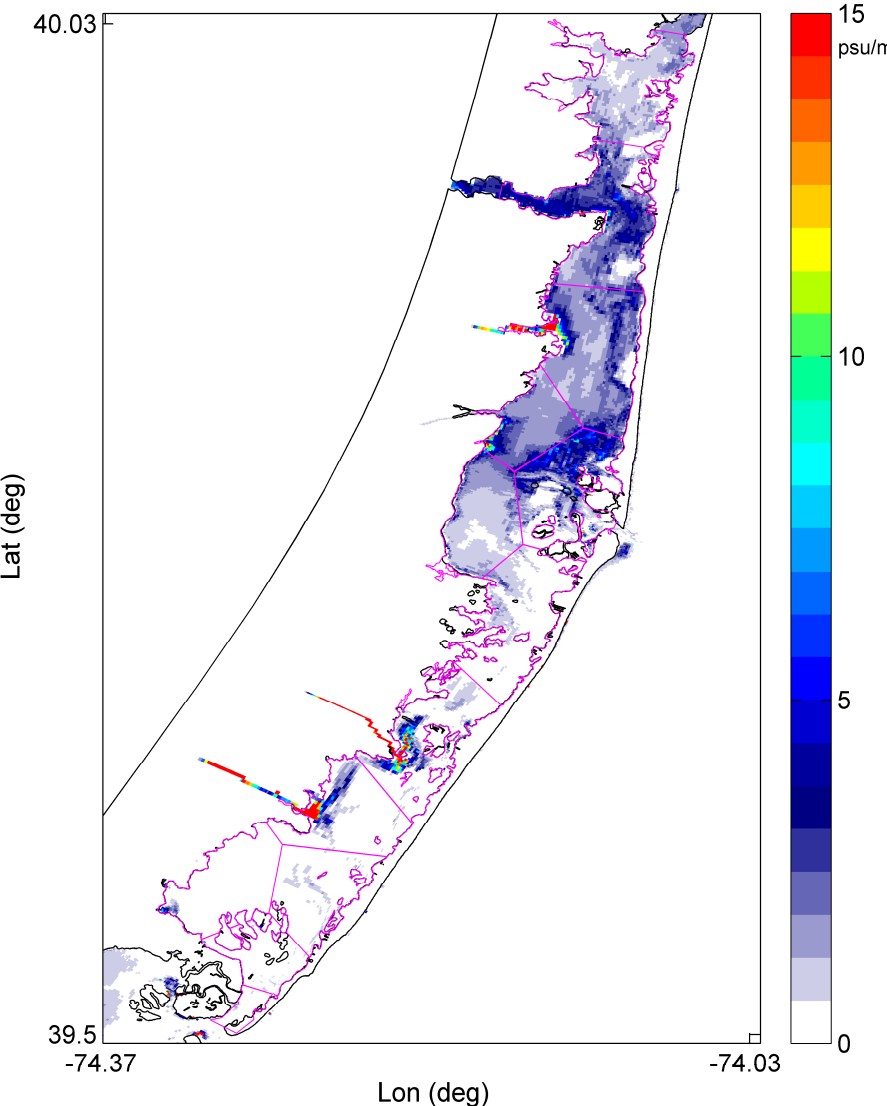

**Figure 3.** Maximum vertical salinity gradient in the water column from the hydrodynamic model over the 22 day simulation period. Gradients above 5 and 1.2 (psu/m) altered the behavior of clam and oyster particles respectively. Magenta lines designate polygons with particle releases occurring at each centroid.

General trends for particle dispersal using neutral, clam, and oyster RCMs were to transport northward in BBLEH (Figure 4), as is evidenced by the larger number of RCM's below the diagonal in the RCM plots from upper-left to lower-right than above it (station order is from north to south). Polygons in the south (polygons 8–11) served as sources for many other polygons in the system (Figure 4). There was limited dispersal for all behavior types from northern polygons (1–7) yielding relatively high local retention (particularly polygons 1–3) (Figure 4). Particle dispersal from south to north were apparent regardless of the number of particles released. An exception was the first 4 days, where some clam and oyster particles dispersed from north to south (Figure 4).

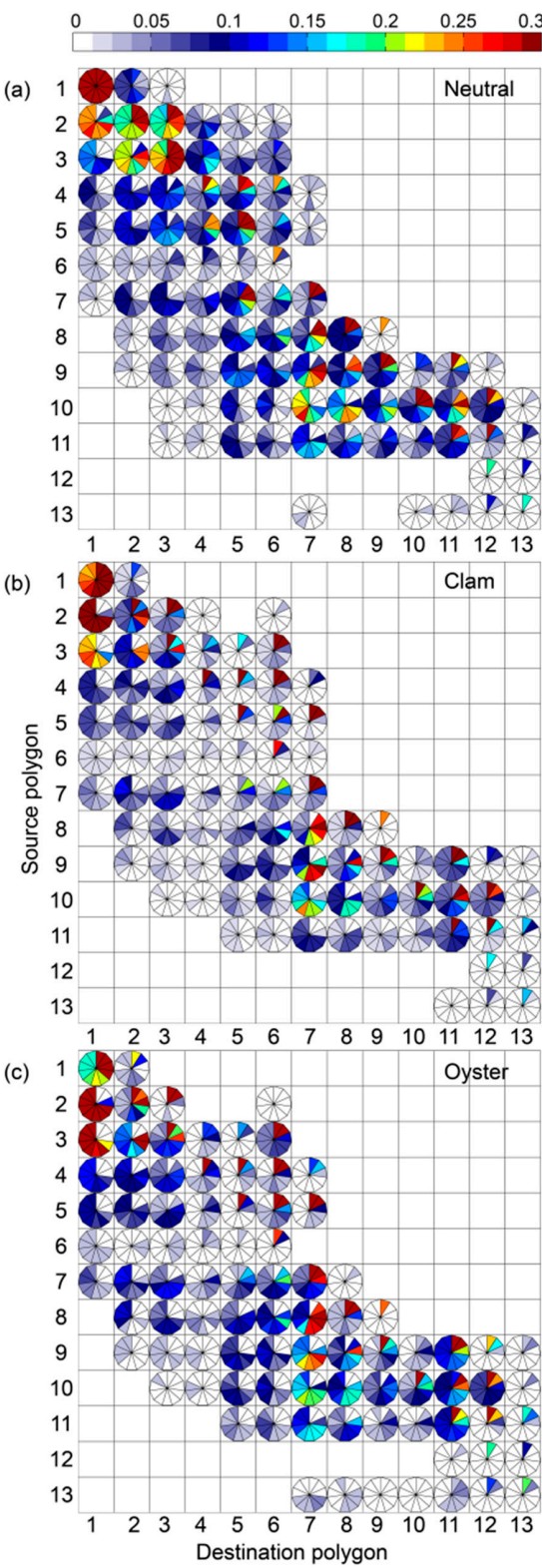

**Figure 4.** Connectivity among release polygons (*y*-axis) and destination polygons (*x*-axis). Color of each 2-day slice shows the proportion (fraction) of particles that move from the source polygon to the destination polygon. The clocks on the diagonal correspond to local retention. All releases use 1000 particles from each release point (n = 13,000 total particles per simulation), panels show connectivity for (**a**) neutral, (**b**) clam, and (**c**) oyster behavior scenarios. Source–destination combinations without a clock (grey box) denote combinations for which no connectivity occurs.

The connectivity for neutral particles in simulations releasing 1000, 2500, and 5000 particles were released per polygon did not vary by more than 5.1% (i.e., the $DRCM_{max}$ for all comparisons did not exceed 5.1%). The $DRCM_{5000-1000}$ (Figure 5), was similar to $DRCM_{5000-2500}$ and $DRCM_{2500-1000}$. This suggests that simulations using 1000 particles represent particle retention and dispersal trends adequately relative to model releases using 5000 particles.

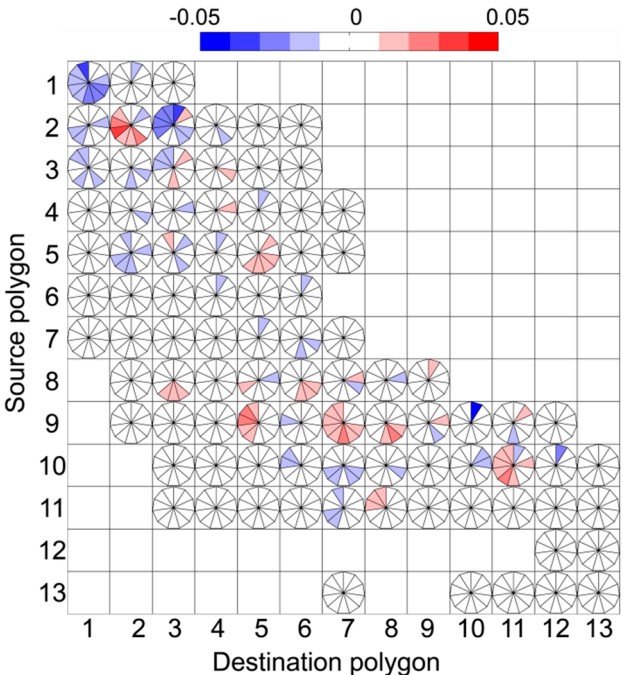

**Figure 5.** $DRCM_{5000-1000}$ showing the change in connectivity when 5000 particles instead of 1000 are released. Positive (red) and negative (blue) values indicate stronger and weaker connectivity differences between RCM model where n = 1000 particles subtracted from RCM model output where n = 5000 particles.

Differences were observed between clam and oyster RCMs (Figure 4). $DRCM_{clam-neutral}$ plot showed retention and dispersal differences of up to 72% in some areas but smaller (<10%) differences overall (Figure 6a). These trends were almost identical to the $DRCM_{oyster-neutral}$ plot (Figure 6b). Greater connectivity (>30%) from polygons 2 and 3 occurred toward polygon 1 when oyster larval behavior was added (Figure 6). Based on these observations, large differences in connectivity were apparent in the north based on particle behavior. Most DRCM differences for larval durations of interest (after 8 days) were not larger than 10% (Figure 6). In addition, when comparing behavior vs. neutral connectivity, positive connectivity differences generally occurred at polygons more distant from the source and negative connectivity differences generally occurred with local retention (i.e., behavior led to longer dispersal distances).

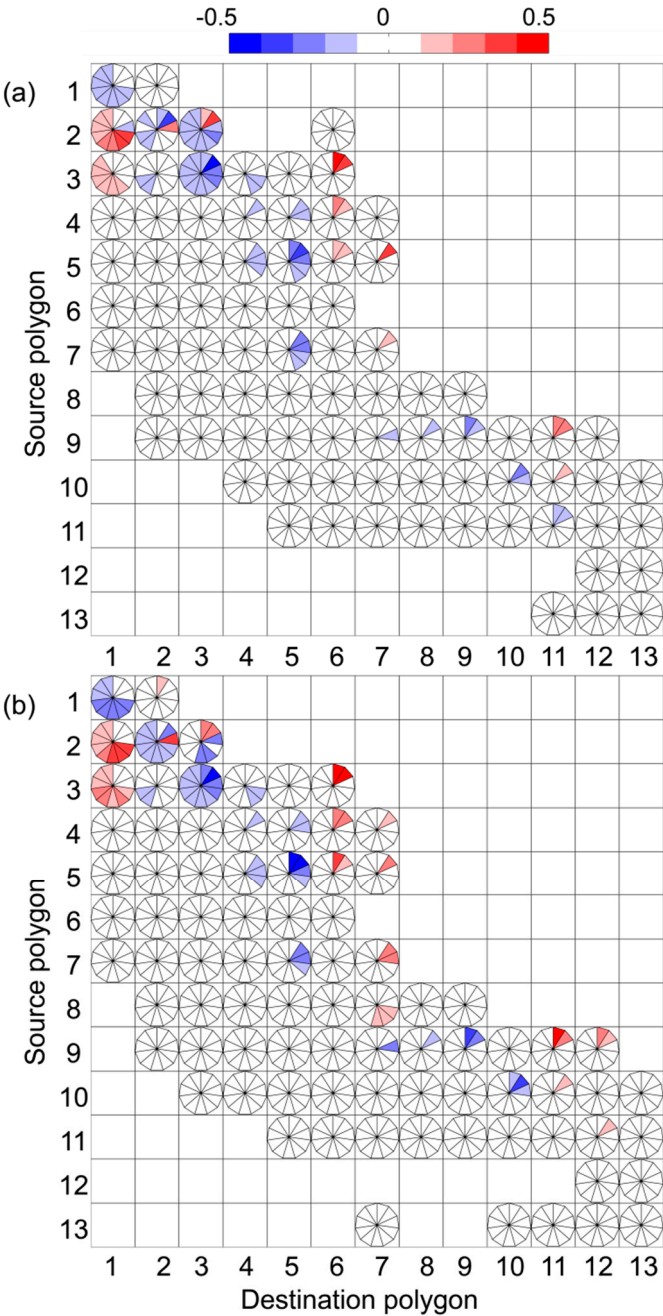

**Figure 6.** (**a**) DRCM$_{clam-nuetral}$ showing the change in connectivity when clam behavior is used instead of neutral particles, and (**b**) DRCM$_{oyster-nuetral}$ when oyster behavior is used. Positive and negative values indicate positive (red) and negative (blue) connectivity differences between particles with behavior (clam or oyster) subtracted from particles with neutral behavior.

The polygons in the traditional connectivity matrix 1–13 were listed from north to south and showed the same general trends as the RCMs (Table 2). Polygons 8 through 11 show the greatest number of connections to other polygons, whereas polygons 12 and 13 had the lowest connectivity to other parts of the system. These traditional connectivity matrix results are similar to what the RCMs display, but do not have the additional details of time added.

**Table 2.** The number of other polygons that a given polygon (left column) contributed 1% or more of its particles to for clam behavior after 8 days (column 2) and oyster behavior after 14 days (column 3). Darker colors indicate that polygon dispersed particles to a greater number of polygons.

| Polygon | Clam | Oyster |
|---------|------|--------|
| 1 | 2 | 2 |
| 2 | 5 | 4 |
| 3 | 6 | 6 |
| 4 | 7 | 7 |
| 5 | 7 | 7 |
| 6 | 7 | 6 |
| 7 | 7 | 8 |
| 8 | 8 | 8 |
| 9 | 10 | 11 |
| 10 | 9 | 10 |
| 11 | 9 | 9 |
| 12 | 2 | 3 |
| 13 | 4 | 7 |

## 4. Discussion

Commercially important shellfish have been in decline in the northeastern United States and some of this decline has been correlated to decadal shifts in climate (specifically the North Atlantic Oscillation) [15]. This work shows that physical parameters influenced by climate patterns (e.g., sea level, current, winds) can have a large impact on larval connectivity Understanding the larval stage provides insight into shellfish population connectivity that can help strategically identify areas for restoration. This modeling study has shown that southern populations of oysters and clams in BBLEH could serve as a larval source for populations to the north based on modeled transport patterns and that those patterns could be influenced by tides, wind, and to a lesser extent larval behavior. The DRCMs showed differences between neutral particles and those with behavior in northern regions where stratification was higher. Increased stratification led to greater differences in transport and connectivity for particles with behavior. The results of this study are consistent with previous studies, demonstrating that while physical transport was the dominant influence for larval dispersal in a shallow system (e.g., [8]), in some situations larval behavior can affect larval transport and ultimately connectivity [12–14]. Managers/restoration practitioners would likely need to combine habitat restoration, stock enhancement, and broodstock protection (i.e., mpa) at different places if, as in this system, transport is generally unidirectional whereby the south provided the north with larvae. Therefore, habitat restoration (for settlement; particularly for oysters) could be the focus in north/north central, habitat restoration, and broodstock protection be the focus in south central, and habitat restoration, stock enhancement and broodstock protection could be the focus in the south (except for polygons 12 & 13).

During the study period, physical processes appeared to influence connectivity more than numbers of particles or particle behavior. The general pattern of connectivity did not vary substantially between neutral, clam, and oyster simulations (Figures 4–6). After 4 days of simulation, the majority of particle movement generally occurred from south to north in the system for all behavior types. One exception to this general trend should be noted when particles traveled from north to south (e.g., Figure 4a–c), specifically at the source destination pairs of (s2d3), (s3d6), (s4d5), (s4d6), (s5d6), (s5d7), (s9d12), (s10d11), (s10d12), and (s11d12) in all RCMs (Figure 4). This north to south transport, during the first few days of the model, may be due to wind (from the north), pushing particles to the south in the first 2 days of the simulation (Figure S1). This wind direction is not typical of southern winds during spawning in this area but did push 2-day-old particles from north to south. Wind patterns have been associated with variations of larval dispersal patterns in other studies of bivalve larval connectivity [12]. Previous research suggested that subtidal water levels and currents in BBLEH were

mainly controlled by coastal sea levels and to a lesser extent by local winds [39]. Overall, our modeling results also demonstrate the relative dominance of coastal sea levels over local winds.

As demonstrated in other larval transport simulations particle transport in these simulations was also affected by behavior [12,13]. In this case, behavior enhanced the transport of particles in the northern polygons and polygons 2 and 3 transported > 30% more particles to polygon 1 when particle behavior was included compared to neutral conditions. From an overall system perspective, movement from polygons 2 and 3 to 1 contributed to retention. After 4 days, simulated oyster particles had greater connectivity in the north for polygons 2 and 3. The clam behavior showed retention and dispersal differences of up to 72% in some areas relative to neutral conditions, but relatively smaller (<10% differences) overall. Likewise, oyster behavior also led to differences in dispersal relative to neutral simulations, some of which varied over time after the first 4 days. In addition, when comparing behavior vs. neutral connectivity, positive connectivity differences generally occurred at more distant polygons (from the source). This suggests that behavior generally led to increased transport as compared to neutral particles in the system. The northward progression of most particles lead to overall system retention. The lack of stratification could explain some of these observations since physical conditions can overwhelm the behavioral responses in larvae in various hydrodynamic environments [40,41].

Stations in the south (with 12 and 13 near the inlet as exceptions) contributed both clam and oyster larvae to most of the other regions, whereas stations in the north were not large sources for areas in the south and generally had a higher rate of particle retention. These results are similar for neutral particles observed in previous simulations [27] and illustrate the physical dominance for much of this shallow system. Particle concentrations over time at the two inlets are low with lower particle retention in polygons near the inlets (specifically 6, 12, and 13). Particles modeled may travel further offshore or to other parts of the coast and represent 23% of the modeled particles potentially being lost but 77% staying within the Barnegat Bay System. Previous studies have shown that inlets can have similar effects on larval transport patterns [42,43].

We found that there is an underlying northward transport in the bay, which at times enhanced by the winds from south. A simulation period of 2 months using the same physical model as this study showed that the general physical patterns of northward progression of neutral particles hold over a wide variety of conditions in BBLEH [27]. The residual current was mostly driven by the tides for the 2-month simulation period (based on a model simulation forced with tides only) [27]. Contribution from tides were more than 75% and northwards with wind contributing 20% of the residual current, but the wind contribution could be negative or positive depending on wind direction. Previous work demonstrated that circulation patterns in the south of BBLEH together with wind action, create a relatively uniform body of water with little stratification in the summer [44]. Therefore, the simulated window of time used in this study is relevant to the system and the species of interest here based on the recorded wind direction during the course of the study period (Figure S1). We simulated connectivity for clams and oysters in the spring because of the availability of a calibrated hydrodynamic model (ROMS) for that period. Clams and oysters spawn during summer time temperature cues [2,18] but, for this study, temperature had no influence in the larval model growth parameter. Temperature could influence the stratification of the shallow system leading to more veliger larvae staying near the surface where they will be transported north by the predominant southerly winds of summer. Extreme patterns in sea level, tidal flow or other physical factors of the system were not modeled under the observed conditions but are possible in some years which could result in different transport patterns than those observed in this study.

Wind influences this shallow estuary and general trends are variable between seasons with south-southwest during summers and west-northwest in the spring [45]. During our modeled conditions, wind direction was mostly from the south with some northern winds toward the end of the modeled period (Figure S1). A trend of southern summer winds would likely contribute to more particles moving northward in this system. Future modeling studies will be important for

understanding overall larval transport patterns that may be influenced by variable physical factors (e.g., sea level, precipitation) that result from climate variation (e.g., the NAO).

Eastern oyster larval ecology and swimming behavior has been well studied [2,9,46,47], but field-based hard clam larval ecology studies are less common [18]. For pediveliger oysters, controlled laboratory studies indicate that smaller scale turbulence and waves affect swimming behavior and can result in "diving" behavior [47–49]. These small-scale behavioral responses were not part of the suite of behaviors for our modeled particles, and future studies could benefit by inclusion of smaller scale physical and other general settlement cues for the biological organism being modeled. Furthermore, studies aimed at understanding how robust larval behavior studies are for modeled algorithms will help future studies characterize model sensitivity.

Larval durations for bivalves, including clams and oysters, can vary based on external factors, including temperature and salinity [2,18]. The clam and oyster larval particles in this study were allowed to disperse for a 22-day period. In most years, clam larvae probably settle within 8 days [18] and oyster larvae within 14 days [2]. Displaying results of RCMs provides observations of larval transport over multiple periods of time. However, future studies should incorporate more complex larval behavior, as these behaviors may further affect dispersal patterns.

Simulations were robust at particle numbers beyond 1000 (13,000 system-wide), with no additional benefit resulting from additional particles (32,500 and 65,000). Releasing 1000 particles from each station (n = 13,000) revealed similar, within 5.1% for transport and retention, results to releasing 5000 particles (n = 5000). This trend was apparent for all behavior types and has implications for the amount of time needed to model this system. In our case, time savings of 4 days for each run were achieved by running 1000 particles instead of 5000.

Unlike typical two-dimensional connectivity matrices, the RCM and DRCM techniques used in this study allow examination of particles at multiple points of time and space. Retention clock matrices can be used to assess time-dependent connectivity between the polygons of the system, and over a range of larval durations. Specifically, RCMs were used to analyze the results from different simulation scenarios (neutral particles, particles with clam and oyster behaviors, and simulations with different numbers of particles). These analyses can provide insight into connectivity for each species case, and differences among species were quantified using the newly defined difference RCM (DRCM) method. These results demonstrate that RCM and DRCM modeling tools could be employed to help understand transport differences (both physical factors and behaviors) of marine larvae for coastal areas world-wide.

**Supplementary Materials:** The following are available online at http://www.mdpi.com/2077-1312/7/6/167/s1, Figure S1: Time series for modeled wind near the center of the computational domain. Wind speeds up to 5 m/s were apparent during March 15–19, which contributed to north-south movement of particles in the system during this period, Figure S2: Hydrodynamic model domain (in magenta) and the bathymetry of the study area. U.S. Geological Survey streamflow gages (orange squares) are labeled by station number [29].

**Author Contributions:** Conceptualization of the project D.M.M., J.D.G., Z.D., J.V. and N.K.G., methodology D.M.M., J.D.G. and Z.D.; software and formal analysis, J.D.G. and Z.D.; data curation, Z.D. and N.K.G.; writing—J.D.G. draft preparation, J.D.G. and Z.D.; writing—review and editing, J.D.G., D.M., Z.D., N.K.G. and J.V.; funding acquisition, D.M.M.

**Funding:** Provided by the Barnegat Bay Partnership. D.M. was partially supported by the USDA National Institute of Food and Agriculture Hatch project accession number 1002345 through the New Jersey Agricultural Experiment Station, Hatch projects NJ32115.

**Acknowledgments:** This work is supported by the Barnegat Bay Partnership EPA grants CE98212311, CE98212312. We extend our deep thanks to anonymous reviewers and Lisa Lucas who provided thoughtful input that improved the manuscript. We thank Matthew Kozak and Ian Mitchell for technical advice and Elizabeth North for LTRANS guidance. Joe Caracapa and Jennifer Gius provided help running remote simulations. COAST model source code is available at https://code.usgs.gov/coawstmodel/COAWST [50]. The hydrodynamic model output is available at: http://geoport.whoi.edu/thredds/catalog/clay/usgs/users/zdefne/GRL/catalog.html [21] and particle tracking model outputs are available from the corresponding author upon request.

**Conflicts of Interest:** The authors declare no conflict of interest. The funders had no role in the collection, analysis or interpretation of data.

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
