# Peer review of "Estimating Connectivity of Hard Clam (Mercenaria mercenaria) and Eastern Oyster (Crassostrea virginica) Larvae in Barnegat Bay"

_jmse, doi:10.3390/jmse7060167_

Round 1

Reviewer 1 Report

Estimating connectivity of hard clam (Mercenaria mercenaria) and eastern oyster (Crassostrea virginica) larvae in Barnegat Bay

Abstract:

·     Since the RCM is the key point of the paper, it would be good to know what it is exactly in the abstract. You then say that the DRCM is small and infrequent, compared with the general northward transport – so I’m left a bit confused what the main findings are. More clarification is needed I think – and perhaps save space by removing the sensitivity results to particle density.

Introduction:

·     No reference: Pucket et al. 2018

·     I think its important here to discuss the other uncertainties in particle tracking, as well as uncertainty to PLD. E.g. larval Spawning/ behaviour/ settlement patterns, seasonal/interannual variability in circulation, model accuracy, PTM accuracy/parameterisation (e.g. interactions of particles with boundaries), etc

Methods:

·     A more detailed description of the ROMS model setup (i.e. forcing data – simply saying the model was forced with winds, tides and freshwater is not enough I think) and validation results is needed. It is important to know the uncertainty levels associated with the physics, before behaviour-driven particles are added. E.g. How well does the model represent the tide-driven currents, and also the weather-driven currents?

·     It would be good to know the level of variability expected from simulations during different periods. You say the “southerly wind conditions [were] similar to those commonly seen during a summer spawning period”, but the reader needs to know how this may vary for different (plausible) release periods of weather conditions.

·     Since PTM trajectories are strongly contingent on behaviour, it is important to know: (a) how robust the behaviour studies are that the PTM algorithms are based on; and (b) how sensitive the modelled outputs are to these behaviour traits. This second part may involve additional sensitivity around certain parameters (e.g. swim speeds, stage periods, sensory cues) that will in themselves show interesting results.

·     Line 203: Repetition of justification of simulation period

·     Line 221: The 1% connectivity threshold needs to be justified. E.g. 0.9% of millions of larva is still a lot of larva!

Results:

·     Fig. 3: Not clear (from this figure and caption alone) what the red line refer to. Better to delete these and add the salinity gradient thresholds as contour lines.

·     Also good to plot residual currents during the simulation period – this will give a lot of information not present as yet, especially since the physics seem to be a strong control on dispersal.

·     Do many particles leave the region and travel further along the coast or offshore? And would this region be a sink for other shellfisheries? Good to infer this from your results.

Discussion:

·     You have rightly discussed and explained the variability caused by your experimental design – and importantly the likely variability that you did not consider. But I am left unconvinced that your snap-shot simulation captures enough of the variability in dispersal and connectivity. For instance, are populations connected only by normal circulation patterns, or do extreme patterns play a role? Getting access to further ROMS simulations and ruining sensitivity scenarios would greatly improve the analysis.

Author Response

Thank you for your careful review. We believe that by addressing your comments we have strengthened our manuscript. 

Reviewer 2 Report

The authors conducted a solid study that should provide good information for resource managers in the region. The study is well presented, overall. My concern is that in many instances the study is not put into a broader context that would make it more widely applicable. For instance, their study area is not the only location to suffer shellfish losses, so their information might be useful for other locations. More specific comments are below.

Line 12 – should be “Many marine organisms…” because in contrast to the species of focus in this manuscript, many marine organisms are not sessile at any point in their life stage.

Line 42 – the juvenile and adult stages are sessile, period. Not relative to the larval stage. This is awkward wording. Maybe just end the sentence after “sessile”.  Earlier in the sentence you’ve already defined the larval stage as planktonic. Perhaps instead of using “transient”, use “dispersive” instead earlier in the sentence.

Paragraph beginning Line 51 – if I understand correctly, there are no official data on commercial shellfish landings, but fisheries independent sampling by NJDEP indicate declining populations? I’m amazed that in 2019 a state doesn’t have data on a commercial fishery. Given that the authors intend their study to have broader scientific implications, I suggest they cite the declines in shellfish landings and populations in other areas to demonstrate that the issue they are addressing has broader implications. Otherwise, only those interested in their study area would have interest in this study.

Line 63 – in addition to restoration and enhancement, isn’t larval dispersal information also important for prioritizing protections of existing shellfish beds?

Line 87 – this is the first time DRCM has been introduced as an acronym. It needs to be described.

Figure 2 – In the figure I was given, the polygons are outlined in red, not white. In the text, can you explain why the polygons are the sizes and shapes? They appear to encompass different amounts of area. Is the likelihood of the presence of clams and oysters similar in all polygons?

Line 165 – “Particles between day 0.5 and day 1.5 were randomly assigned upward or downward swimming at each time step with a high probability (90%) of swimming upward.” Larvae of many species exhibit predictable diel vertical migrations. Based on the information provided here, it appears that larvae of clams and oysters do not? Or does the salinity gradient negate that type of vertical migration? (Same comment on sentence that begins on Line 189.) This was addressed to some extent in the Discussion, but may be useful to at least mention earlier to better justify the methods.

Paragraph Line 248 – why were these PLDs of 8 and 14 days chosen? Are these mean, median, mode for PLD for these species? What is the justification for these PLDs for this analysis?

Line 260 – “Within the first four days of simulation, winds were blowing from the northeast with speeds up to 5 meters per second (Fig. S1).” But previously (Line 119), it was stated that although the available physical model output was from spring (prior to the normal summer spawning period), a time period with south winds was used for this model simulation. Please clarify. Are northeast winds common in summer months or is this an anomaly? This is addressed somewhat in the Discussion (beginning Line 336), but is not addressed in detail. Since wind forcing is likely a major factor in such a shallow estuary with limited tidal connections, I think this deserves more attention. Was the wind direction an anomaly? If not, then this might be a significant finding rather than an “exception”. (Line 336 – 10 instances is not “one exception” to the general trend.) If it is an anomaly and not expected to be a regular occurrence, then clearly state this. Why weren’t all larvae transported southward during this northeast wind event?

Line 310 – change “This” to “These”

Paragraph beginning Line 318 – what are/were the causes of clam and oyster population declines in the study area? Do the suggestions for resource managers in the paragraph address the causes (e.g., loss of oyster reef settlement habitat) or are the causes associated with water quality or other issues? Given the importance of the salinity gradient to larval behavior and transport, if a cause of shellfish declines was a change in freshwater flows into the estuary, this would be important information. Knowing this would help place the conclusions and management applications in context and address the applicability of the study findings.

Author Response

Thank you for your comments. We have addressed your comments and believe the manuscript has benefited. 
